# Orally Administered Amphotericin B Nanoformulations: Physical Properties of Nanoparticle Carriers on Bioavailability and Clinical Relevance

**DOI:** 10.3390/pharmaceutics14091823

**Published:** 2022-08-30

**Authors:** Shadreen Fairuz, Rajesh Sreedharan Nair, Nashiru Billa

**Affiliations:** 1School of Science, Monash University Malaysia, Jalan Lagoon Selatan, Bandar Sunway, Subang Jaya 47500, Selangor, Malaysia; 2School of Pharmacy, Monash University Malaysia, Jalan Lagoon Selatan, Bandar Sunway, Subang Jaya 47500, Selangor, Malaysia; 3College of Pharmacy, QU Health, Qatar University, Doha 2713, Qatar; 4Biomedical and Pharmaceutical Research Unit, QU Health, Qatar University, Doha 2713, Qatar

**Keywords:** oral, amphotericin B, bioavailability, nanoparticle, clinical

## Abstract

Amphotericin B is an effective polyene antifungal considered as a “gold standard” in the management of fungal infections. Currently, it is administered mainly by IV due to poor aqueous solubility, which precludes its delivery orally. Paradoxically, IV administration is akin to side effects that have not been fully eliminated even with more recent IV formulations. Thus, the need for alternative formulations/route of administration for amphotericin B remains crucial. The oral route offers the possibility of delivering amphotericin B systemically and with diminished side effects; however, enterocyte permeation remains a constraint. Cellular phagocytosis of submicron particles can be used to courier encapsulated drugs. In this regard, nanoparticulate delivery systems have received much attention in the past decade. This review examines the trajectory of orally delivered amphotericin B and discusses key physical factors of nanoformulations that impact bioavailability. The review also explores obstacles that remain and gives a window into the possibility of realizing an oral nanoformulation of amphotericin B in the near future.

## 1. Introduction

Patent expiry of proprietor drugs provide opportunities for pharmaceutical scientists and manufacturers alike to explore formulation modalities that potentiate the therapeutic efficacy of the drug further. With the advent of nanotechnology and related technologies, drugs that were considered “problematic” with regard to poor systemic bioavailability imposed by low solubility and/or permeability have emerged as prime candidates worthy of inclusion in the formulation lineup of novel drug delivery systems. One such drug candidate is amphotericin B (AmpB), a polyene antifungal agent that is very effective against several strains of fungal infections [1]. AmpB is a class IV drug according to the BCS classification system, exhibiting low solubility in neutral pH and low membrane permeability. Obtained from cultural media of bacteria *Streptomyces nodosus*, AmpB has a molecular weight of 924.1 g/mol and is fungistatic and/or fungicidal depending on the dose employed. It is effective against most human-infection-causing fungi except some species such as *Candida lusitaniae*, *Trichosporon* spp., *Geotrichum* spp., and *Scedosporium apiospermum* [2]. AmpB is also effective against some protozoal infections such as leishmaniasis and trypanosomiasis, caused by *Leishmania* and *Trypanosoma cruzi*, respectively [3]. *Leishmania* is an intracellular parasite that mostly accumulates in liver, spleen, and bone marrow; hence, Amp B has to be formulated in order to successfully reach these target sites [4]. For systemic fungal infection, 0.5–1.5 mg/kg of the drug is administered as slow intravenous infusion (IV) over 4 to 6 h, dictated by the severity of the infection, type of the pathogen, kidney function, and the tolerance level of the patient [2,5]. For targeting the liver via portal blood circulation, the oral route is ideal, while lymphatic uptake can enable bone marrow targeting [4]. AmpB’s plasma half-life is typically within 24–48 h, engaged in three-compartmental distribution kinetics with an elimination half-life of 15 days [2,5]. This suggests slow tissue assimilation of AmpB from the peripheral compartment. AmpB exemplifies the type of drug that warrants formulation intervention in order to harness its full therapeutic potential. Therapeutically, AmpB works by associating with ergosterol found in leishmanial and fungal cells or cholesterol (mostly in mammalian cells), creating pores that allow drainage of essential nutrients such as K^+^, ultimately leading to cell death [5]. Crucially, AmpB has a higher affinity for ergosterol than cholesterol, so there is a degree of selectivity to leishmanial and fungal cells compared to mammalian hosts. However, as accumulation in tissue increases, its affinity for cholesterol becomes significant, resulting in severe toxicity to that tissue or organ [6]. Despite being in existence for over half a century, AmpB continues to be pivotal in clinical use, with excellent in vitro activity against a broad range of fungal infections, including *Candida*, *Aspergillus*, and *Mucorales* spp. [7]. Very early studies on AmpB oral formulations presented promising results when a colloidal suspension of AmpB comprising *N*-dimethylacetamide and hydrochloric acid was orally administered to mice [8]. AmpB doses of up 40 mg per mouse resulted in a remarkable 90–100% survival rate in *Candida-albicans*-infected mice. Other early and related studies on mice showed promising results from AmpB administered orally [9]. However, when translated to humans, the results were disappointing even on IV administration [10]. Other routes of administration have also been explored, including intrathoracic and intra-articular for effectiveness against leishmanial and fungal infections, and the conclusion drawn from these studies is that the most practical route for the delivery of AmpB is IV [11]. Thus, AmpB is currently approved for use in a number of IV formulations; for example, AmpB solubilized by sodium deoxycholate is considered as the conventional formulation. Its use is limited due to toxicity concerns. Furthermore, due to its extreme lipophilicity, newer-generation lipid-based carrier systems, which are less lipophilic, have emerged. AmpB lipid complex (ABLC) comprises two lipids: _L_-a-dimyristol phosphotidylcholine (DMPC) and _L_-a-dimyristol phosphotidylglycerol (DMPG). DMPC and DMPG are present in the molar ratio of 7:3, which in turn is a 1:1 molar ratio with AmpB [12]. A colloidal dispersion (ABCD) is a 1:1 ratio of cholesteryl sulphate and AmpB [13]. All these intravenous formulations have served well clinically insofar as the alleviation of some systemic fungal infections are concerned. Nonetheless, patients have to contend with some very severe side effects. Due to its amphipathic nature, AmpB interacts with water to form aggregates along with the monomers, whereby systemic toxicity is related to release of potassium ions and/or dissociation of membrane and, in the case of red blood cells, release of hemoglobin [3]. The monomer is more water-soluble and less toxic than the aggregates. High concentrations of AmpB promotes aggregation [3]. There is the suggestion that the “super-aggregated” state of AmpB is less toxic than the loosely aggregated form [3].

AmpB IV infusions have commonly been criticized for causing nephrotoxicity, as evidenced by decreased glomerular filtration rate and tubular dysfunction. Its slow excretion from the body also means that a low dose of the drug has to be prescribed. When given via the IV route, AmpB tends to interact with cholesterol from tubular cell membranes, leading to damage, and the drug’s vasoconstricting function in the kidney reduces the glomerular filtration rate, with both mechanisms often being interrelated and affecting each other [14]. Other infusion-related side effects include fever, shaking chills, hypotension, anorexia, nausea, vomiting, and headache, which often require infusion of the drug to be slower. It may be followed by adding aspirin, antipyretics (acetaminophen), antihistamines, or antiemetics to the treatment to prevent changes to the infusion rate [15]. On the other hand, low solubility in physiological pH, poor absorption, and hence low bioavailability of the drug also remain a pressing concerns that needs to be addressed when designing an oral formulation [16].

Recent advances in formulation technology have reignited the quest for viable oral formulations of poorly soluble/permeable drugs. In particular, nanotechnological formulation techniques capable of modifying the physicochemical properties of the drug and whereby the solubility is not implicated during absorption seem to be a rational approach. In the case of AmpB, shielding from the acidic environment of the gastrointestinal tract is pivotal [17].

The gastrointestinal tract has the propensity of anchoring submicron particles, enabling subsequent uptake. This provides a platform for the possible administration of AmpB designed as nanoformulations. In this regard, nanosuspensions of AmpB in an aqueous solution comprising Tween 80, Pluronic F68, and sodium cholate have been shown to adhere to the gastric mucosa with the consequence of increased uptake in mice [18]. Treatment of the above formulation on Balb/c mice infected with *L. donovani* showed a reduction in parasites compared to micronized AmpB, Fungizone, or AmBisome. AmpB dissolved in methanol mixed with lipids comprising glyceryl tristearate and/or glyceryl monostearate stabilized with phosphatidylcholine showed a significant increase in serum levels in rats compared to AmpB suspension in aqueous medium [19]. AmpB in chochelate nanoformulation tested on mice also showed a dose-dependent reduction in fungal infection [20]. These with several other nanoformulations of AmpB described in the literature are indicative of the possibility of the emergence of viable oral nanoformulations of AmpB. Oral nanoformulations that ensure the monomeric form of AmpB is unavailable for binding with cholesterol or ergosterol in the plasma membrane will address the toxicity concerns akin with current AmpB IV formulations. Yet, there are no viable oral AmpB nanoformulations on the market. This review investigates the applications of nanotechnology in the development of oral AmpB formulations spanning the last two decades and probes the trajectory towards the possible realization of oral AmpB nanoformulations suitable for human clinical use. Articles published in English between 2000 to 2021 related to amphotericin B nanoformulations were retrieved using *Google Scholar*, *ProQuest*, *ScienceDirect*, *Wiley*, and *PubMed*. The mesh keywords such as oral, amphotericin B, nanoformulation, nanoparticles, liposomes, polymeric nanoparticles, micelles, solid lipid nanoparticles (SLN), nanostructured lipid carriers (NLC), polymer–lipid hybrid nanoparticles (PLN), and self-emulsifying drug delivery systems (SEDDS) were used as the input in the databases.

## 2. Overview of AmpB Nanoformulations

The quest for improved bioavailability and tolerability after administration of AmpB has led to the proposal of several nanoformulations, some of which are discussed here and shown in Figure 1. AmpB encapsulated in nanoparticles intended for oral delivery comprising of natural polymers such as chitosan have particularly become popular because of its biocompatibility, biodegradability, and low cost [21]. Furthermore, chitosan is also mucoadhesive, which favors pre-anchoring to cellular mucosa prior to uptake [22]. Synthetic polymers such as poly(lactic acid) (PLA) and its copolymer poly(lactic-co-glycolic acid) PLGA have also been used to encapsulate AmpB in nanoformulation with varying degrees of success [23]. The release of AmpB from polymeric nanocarriers is mostly preceded by desorption or diffusion from the polymer coating or matrix, respectively [24].

One of the key strategies for controlling the release of drug molecules is to alter the pore size of nanocarriers. Triggering factors such as temperature and pH can also affect the internal conformation and permeability of a drug delivery system to provide controlled drug release. Sterically stabilized nanocarriers containing block co-polymers such as poly lactic acid-b-PEG-folate or poly histidine-b-PEG-folate will undergo change in conformation in response to pH variations. Alternatively, polymeric vesicles and nanoemulsions utilize ultrasound to control the drug release by stimulating the nanocarrier components [25]. Drug release from some polymer nanocarrier is preceded by polymer erosion [25]. In this regard, the molecular weight of the polymer and density of the polymer matrix have been implicated to modulate release, where higher molecular weight or density lead to lower drug release rates [25]. Polymer nanoparticles are simpler in configuration, with better stability profiles compared to other nanoformulations. They are also more stable in biological fluids [24]. Co-polymers made of ε-caprolactone (CL), trimethylenecarbonate (TMC), and mmePEG_750_ form micelles when gently mixed in aqueous media, entrapping poorly soluble drugs such as Amp B [26]. Free AmpB has a solubility of about 100 μg/mL in pH 2–11 [27], but this can be improved to more than 250 μg/mL when formulated in poly(ethylene oxide)-block-poly(*N*-hexyl stearate-l-aspartamide) micelles, with a yield of up to 77% [26]. Micelles self-assemble to form amphiphilic molecules with the hydrophilic regions oriented towards aqueous media and forming a hydrophobic core. Within this architecture, it is possible to encapsulate poorly soluble drugs such as AmpB within the core. When encapsulated in micellar structures, AmpB is believed to exist in the monomeric and therefore has a lesser chance of causing hemolysis of blood cells [28]. Micellar encapsulation also improves the chemical and physical stability of AmpB and improves its distribution within tissue. However, there are also reports of decreased antifungal activity of AmpB in polymeric micelles, for example, poly (ethylene glycol)-block-poly(ε-caprolactone-*co*-trimethylenecarbonate) polymeric micelles [26]. In the work by Seranno et al. [3], the uptake of AmpB encapsulated in *N*-palmitoyl-N-methyl-*N*,*N*-dimethyl-*N*,*N*,*N*-trimethyl-6-O-glycol chitosan (GCPQ) nanoparticles following oral administration achieved a bioavailability of 24%. Furthermore, comparable efficacy to parenteral liposomal AmB (AmBisome) was demonstrated in disease models.

There is currently a move towards utilizing lipid-based nanoformulations to courier AmpB orally due to reduced toxicity. For example, liposomal nanovesicles comprised of hydrogenated soya phosphatidylcholine, cholesterol, and distearoylphosphatidylglycerol at a ratio of 10:5:4 have been successfully used to encapsulate AmpB, resulting in liposomes of about 80 nm diameter, whereby higher plasma concentrations were achieved compared to unencapsulated AmpB following oral administration [2]. Other lipid nanoformulations have achieved a long plasma half-life of up to 152 h and lower clearance than conventional AmpB deoxycholate intravenous infusions, which is crucial in long-term therapy [29]. Liposomal AmpB is less nephrotoxic than AmpB deoxycholate owing to preferential distribution in the spleen and liver; thus, there is less free drug interaction with distal renal tubules and lower glomerular filtration due to the small size of the liposome. Most of the lipid bilayers in liposomes comprise of hydrogenated phosphatidyl choline and relatively stable. The fatty acid chain that complexes with the amine group of AmpB is negatively charged. This ensures retention of AmpB within the core of the liposome. Similarly, cholesterol in the liposomal bilayer attaches to and holds the drug in the vesicles [29].

More recently, solid lipid nanoparticles (SLN) have emerged as one of the frontrunners for the oral delivery of AmpB. AmbiOnp is an oral AmpB formulation that presents low levels of AmpB distribution in the kidney compared to IV delivery of Fungizone [30]. SLN employs the advantage of the possibility of incorporating biological lipids within the matrices, thus minimizing carrier toxicity, with the possibility of modifying AmpB release as well. The core of the SLN comprises lipids that are solid at room temperature and stabilized by surfactants/emulsifiers. Lipids employed in SLNs are affordable and safer than those used in other lipid nanoparticles [30]. Thus, SLNs have been widely used for studying the oral delivery of AmpB. It can retain the solubilized state of AmpB in gastrointestinal fluids as mixed micelles by inducing the secretion of bile salts and phospholipids [30]. In this regard, the effect of food on the absorption of AmpB is noteworthy and has been inferred to be promoted by uptake via the lymphatic system, making the small intestine the favored site for the delivery of SLNs [24].

Metallic nanoparticles such as gold (Au-NPs) or silver nanoparticles (Ag-NPs) represent another frontier in the quest for efficient oral delivery of AmpB [31,32]. These nanoparticles have become popular in the delivery of drugs due to ease of functionalization with drug moieties or targetable functionalities that facilitate permeation across biological membranes [32]. Au-NPs chemically conjugated with AmpB via a lipoic acid cross-linker presented 5-fold lower IC_50_ against amastigotes, 2.5-fold lower IC_50_ against promastigotes, and better potency against *Candida albicans* compared to free AmpB [32]. The complex also manifests an increased immunostimulatory effect and depletes the ergosterol content of the parasite’s membrane, leading to increased fluidity and loss of cellular content [32].

Other oral AmpB nanoformulations investigated include protein and hybrid nanosystems. For example, an electrospun gelatin nanosystem containing AmpB formed as oral tablets provided a slow release within the gastrointestinal tract [33]. Hybrid AmpB nanoparticles intended for oral delivery have been formulated to comprise polymer and lipid combinations such as a lecithin–gelatin combination reported to improve oral bioavailability of AmpB with a sustained release [34]. 

A summary of investigational AmpB oral nanoformulations spanning the past two decades is presented in Table 1 along with fabrication methods, key features, and pharmacokinetic outcomes.

## 3. Factors Affecting Bioavailability of Orally Administered AmpB Nanoformulations

### 3.1. Impact of Nanoencapsulation of AmpB on Oral Bioavailability of AmpB

Clinical data indicate an increase in the cases of fungal infections within the past few decades, and this is largely attributable to increased prevalence of immunosuppressed patients undergoing long-term therapies or patients who have undergone organ transplants. Other contributing factors include indiscriminate use of broad-spectrum antibiotics or antifungal agents over extended periods, which eventually results in drug resistance [47]. Such trends call for concerted efforts to evolve more effective therapeutic options than current modalities. AmpB is regarded as the “gold standard” for managing fungal infections; however, its poor solubility and membrane permeability result in low bioavailability when administered orally [47]. As mentioned in Section 2, nanoparticulate delivery systems have emerged as formidable drug carrier systems capable of delivering poorly soluble/absorbable drugs systemically when administered orally. They have the smallest size-to-volume ratio of all dosage forms and can be appropriately formulated to modify the physicochemical properties of the drug cargo in favor of improved solubility and hence bioavailability [48]. An improvement in bioavailability can be attributed to the uptake of nanoparticles by absorptive cells followed by drainage in the portal vein. In this regard, lipid nanoparticles have the added advantage of being deployed in the lymphatic system, which avoids the first liver pass. The gastrointestinal tract presents physiological and anatomical constraints that challenge the free entry of poorly soluble drugs across the epithelia. In the case of AmpB, conversion to the monomeric form in gastric pH is an additional constraint when delivered orally. When appropriately encapsulated, AmpB can be shielded from the external milieu and potentially retain its stability prior to uptake [48]. Figure 2 captures possible uptake pathways of orally administered AmpB nanoformulations.

Furthermore, polymeric nanoparticles with attendant mucoadhesive properties can delay the transit of the nanoparticles within the gastrointestinal tract through interactions with the adherent mucous layer mucus longer, generating a gradient for uptake toward the systemic circulation [50]. The stability of AmpB is another key factor related to improved bioavailability from oral administration. In this regard, the protection offered by the carrier system from the external milieu such as extreme acidic pH has been recognized. Conjugation of specific chemical moieties on the surface of the nanoparticle that facilitate internalization is also crucial to ensuring bioavailability. Mannose-anchored thiolated chitosan nanocarriers yielded 6.4 times higher AmpB bioavailability compared to the free AmpB owing to enhancement in intestinal permeation due to the presence of mannose functionality [36].

One contributing factor to decreased bioavailability of orally administered drugs is their susceptibility to efflux proteins such as P-glycoprotein (P-gp), which are highly expressed in the small intestine. This constraint can be circumvented via nanoparticulate encapsulation, with a consequence of improved bioavailability [48]. Through this application, the systemic bioavailability of AmpB loaded in PEGylated polylactic-polyglycolic acid copolymer (PLGA-PEG) nanoparticles following oral administration in rats was 36.4% higher than the colloidal formulation (Fungizone) [23]. The stability of the nanocarrier system within the gastrointestinal tract also correlates with how much cargo is eventually deployed systemically. It has been reported that gelatin lipid hybrid nanoparticles registered a 4.7-fold increase in bioavailability compared to free AmpB in rats [33] and this increase was due to the biocompatibility provided by the lipid and structural integrity of the nanoparticles provided by the polymer in the gastrointestinal tract. In a related study, AmpB encapsulated in phytantriol (PT) liquid crystalline nanoparticles (LCNPs) and glyceryl monooleate (GMO)–liquid crystalline nanoparticles (GLCNPs) showed 6- and 3.5-fold higher oral bioavailability than AmpB in rats, respectively [51]. Here, both LCNPs protected the drug against lipase-mediated GI degradation and P-gp efflux from enterocytes, which eventually led to better absorption [51]. In a study involving AmpB SLN, a 215.6% higher bioavailability than AmpB solution was attributed to the protection offered by the SLN to AmpB from degradation [30]. Thus, nanoparticulate formulations may be employed in the oral delivery of AmpB, with improved bioavailability, and this is borne of the fact that when appropriately encapsulated, the physicochemical properties of AmpB are masked, and uptake across the gastrointestinal epithelia is solely dependent on the nanoparticle. In this context, there are two possible pathways for the uptake of AmpB into enterocytes: namely paracellular and transcellular. Paracellular uptake occurs at the tight junctions between adjourning cells, where uptake is slow and size- and charge-selective. On the other hand, transcellular uptake of nanoparticles occurs through the cells pinocytotically or phagocytotically [38]. Orally administered drugs in most instances passively diffuse across the gastrointestinal epithelium into the systemic circulation. In this regard, the molecular size of drug plays an important role in the rate process [50]. Higher molecular weight poses a constraint to permeability via passive diffusion, and aptly, with a molecular weight of 924.08, AmpB is bound to present a limited capacity for passive diffusion [52]. Uptake of particulates in the order of (400–1000 nm) from the gastrointestinal tract is possible via pinocytosis or phagocytosis by enterocytes. This system of uptake has been explored for the delivery of class IV type drugs of the Biopharmaceutical Classification System (BCS) classification [22] to which AmpB belongs. As in the case of passive diffusion of drug molecules, the particle size of the delivery system is critical to the uptake process and assimilation within the enterocytes cells. In some cases, the nanoparticles remain intact with their cargo and are deployed in the lymphatic system or directly in the systemic circulation [50]. Accordingly, AmpB has been encapsulated in phytantriol liquid crystalline nanoparticles (PLCNP) as well as glyceryl monooleate liquid crystalline nanoparticles (GLCNP), both with sizes less than 210 nm and a yielded oral bioavailability of 6- and 3.5-fold higher than free AmpB, respectively [51]. Furthermore, AmpB encapsulated in gelatin-coated hybrid lipid nanoparticles of size 253 ± 8 nm were about 5-fold higher in oral bioavailability compared to free AmpB [34]. AmpB encapsulated in ethyl cellulose nanoparticles of size 150 ± 9.23 nm presented a 15-fold higher relative bioavailability compared to free AmpB [53]. A 6.4-fold higher relative bioavailability was also reported from mannose-anchored chitosan nanocarriers (482 ± 14 nm) compared to free AmpB [36]. These data suggest that when appropriately encapsulated, we are able to improve the bioavailability following oral administration of AmpB significantly. It is suggested that particles of size range in the order of about 100 nm size generally translate to a higher propensity for uptake by the enterocytes compared to larger sizes, a phenomenon referred to as “particle size-dependent exclusion” [50].

### 3.2. Effect of Encapsulation Efficiency on Oral Bioavailability

We note from the previous sections the various modalities adopted for encapsulating AmpB that are aimed at improving its oral bioavailability. Herein, the amount of AmpB encapsulated within the nanoparticle dictates how much is deployed systemically. Expressed as percentage, the encapsulation efficiency (%EE) refers to the percentage of the cargo within the nanoparticle relative to the total amount of active drug used in formulation [54]. Theoretically, the higher the EE%, the greater the amount of active deployed in systemic circulation in a given time and hence higher bioavailability. However, high nanoparticle EE% does not always correlate within high bioavailability because release of the active is dictated by several parameters. For example, SLN can expel the cargo during storage when the lipid begins to crystallize during storage [55]. Liquid crystal nanoparticles (LCNPs) are able to perturb this conversion to crystalline conformation due to the presence of liquid lipid domain within the crystal matrix of the LCNP, thus restricting the growth of the liquid crystals. This restriction in growth also promotes high retention of cargo within nanoparticles and hence EE%. For example, the EE% of AmpB in phytantriol (PT) and glyceryl monooleate (GMO)–liquid crystalline nanoparticles (LCNPs) both recorded EE% of more than 85% and very high bioavailabilities [51]. AmpB SLN with EE% of 95.09 ± 1.41% also registered very good bioavailability compared to free drug AmpB [30]. TPGS-SLNPs (d-α-tocopheryl polyethylene glycol 1000 succinate-loaded AmpB and Paromomycin) [56] and AmpB-loaded phytantriol cubosomes [57] with high encapsulation efficiencies of 97.99 ± 1.6%, 94 ± 1.5%, and 91.8%, respectively, all have improved bioavailability.

### 3.3. Effect of Surface Modification of Nanoparticles on Oral Bioavailability

Surface modification of AmpB-containing nanoparticles can be used to further either enhance particle uptake due to specific interaction with gastric mucosa or delay their transit time within the gastrointestinal tract, both of which can result in significant mass transfer to the systemic circulation. In a study involving an AmpB-containing nanostructured lipid carrier decorated with chitosan coating (ChiAmpB NLC), enhanced mucoadhesion was observed compared to uncoated NLC (84.2 ± 5.1% versus 55.8 ± 16.1%, respectively) [39]. Even though no in vivo study was conducted, the authors proposed that delayed gastrointestinal transit due to mucoadhesion is likely to result in better uptake and bioavailability when administered orally [22,39]. In another study, nanoparticles comprising stearic acid and conjugated with vitamin B12 were used to encapsulate AmpB, whereby the SLN presented improved bioavailability following oral administration. Vitamin B12, being a natural substrate, is absorbed via an intrinsic factor when attached to the surface of nanoparticles and thus serves to boost uptake of the AmpB SLN. Cellular uptake studies on J774A.1 cells confirmed that uptake was prompted by vitamin B12 moiety of the SLN [58].

Similarly, gum tragacanth (GT)-coated AmpB-containing gold nanoparticles were formulated as a hybrid carrier system, where its mucoadhesivity allowed slow gastrointestinal transit and higher plasma concentration compared to the uncoated AmpB formulation and free drug solution [59]. A d-α-tocopheryl polyethylene glycol 1000 succinate-modified AmpB (TPGS)- and paromomycin (PM)-loaded SLN showed maximal inhibition against *L. donovani*-infected J774A.1 macrophages compared to AmBisome, free AmpB, and PM [56]. Here, surface modification ensured better macrophage targeting and internalization. In a related study, TPGS stabilized PLGA nanoparticles containing AmpB showed 793.2% relative oral bioavailability in contrast to Fungizone [52]. Other reported surface modifications of nanoparticles include the use of Eudragit L30D-coated AmpB-containing nanoparticles, where Eudragit coating protects AmpB from the gastric environment. AmpB in piperidine-coated guar gum nanoparticles, in which piperine, being a natural bioenhancer and guar gum a natural polysaccharide that specifically targets mannose-like receptors on macrophages, also led to improved bioavailability of the nanoformulation [60]. Figure 3 is a pictorial representation of how surface modifications have led to improved bioavailability of AmpB nanoformulations.

### 3.4. Effect of Stability of AmpB Offered by Nanoparticle in Gastrointestinal Fluids on Oral Bioavailability

The low bioavailability of AmpB due to its poor aqueous solubility is further compounded by its poor stability in acidic media. Thus, a viable AmpB oral nanoformulation must be resilient to the degradation effects caused by external media prior to absorption across the epithelia. Even though the amount of AmpB absorbed across the epithelia after premature release from the nanoparticles is minimal, any exposed drug to gastric pH is subject to gastric degradation within the stomach, which means that less cargo is retained within the nanoparticle prior to uptake by the enterocytes [22]. Nanoparticles have shown to protect AmpB from GI degradation so that it can reach its target organ for absorption. They also allow controlled and sustained drug release, which translates to better bioavailability, therapeutic index, solubility, and residence time of the drug. It is often dependent of the nanoparticle’s hydrophobicity, drug release rate, cellular interaction, zeta potential, etc. [22]. 

In this section, we review some of the innovative strategies adopted in AmpB nanoformulation aimed at shielding it from gastrointestinal degradation. AmpB-containing poly (ε-caprolactone) nanoparticles coated with chitosan, for example, were incubated in simulated gastric fluid (SGF, pH 1.2) for 2 h, followed by simulated intestinal fluid (SIF, pH 6.8) for 4 h, which resulted in only 29% degradation, signifying the possible retention of 70% cargo prior to uptake when administered orally [21]. The percentage of AmpB released from a vitamin B_12_–stearic acid conjugate coated on SLNs was only 1.57 ± 0.085% in SGF within 2 h and 7.6 ± 1.48% in SIF within 4 h, and these attributes caused an improvement in the bioavailability [58]. Here, the presence of stearic acid extends the nanoparticle’s stability, as it is not hydrolyzed by pancreatic lipases [58]. An insignificant change in size was observed in AmpB-containing phytantriol and glyceryl monooleate liquid crystalline nanoparticles (PLCNP and GLCNP, respectively) in simulated GI fluids. After 2 h in SGF, both types of LCNPs released less than 5% of AmpB. Furthermore, less than 8% AmpB was released in SIF during 6 h. LCNPs are self-arranged bicontinuous structures with broad solubilization range due to their fluidic and rigid nature that gives better stability to the nanoformulation. Moreover, it was lyophilized using 5% *w*/*v* mannitol, which further improves the stability of the formulation. However, comparing PLCNP and GLCNP, the absence of an ester bond in the phytanyl backbone of phytantriol gives PLCNP better stability than GLCNP. Hence, AmpB-PLCNPs was 6-fold higher in oral bioavailability than free AmpB, while AmpB-GLCNPs 3.5-fold higher than free AmpB [51]. Thus, it proves that nanoparticles often conjugated with a coating tends to improve stability by entrapping the drug in a compact matrix space that is not easy to hydrolyze, thus retaining the drug more until it is in the target site [21].

## 4. Clinical Trials Involving Oral AmpB Nanoformulations

Although several oral AmpB nanoformulations have shown promising in vitro and animal study data, there is still some work to do before successful candidates emerge. Martina Biopharma Inc, USA, has conducted a phase I clinical trial investigating tolerability and oral pharmacokinetics of cochleate-containing AmpB (CAmpB) suspensions in healthy human subjects with some promising results [61]. A single-dose, double-blind study comprising three cohorts (16/cohort) received 200, 400, and 800 mg oral doses of CAmpB for 2 weeks. All groups experienced gastrointestinal adverse effects, such as abdominal pain, aerophagia, diarrhea, nausea, and vomiting, at 6, 38, and 56%, corresponding to 200, 400, and 800 mg doses, but symptoms were mild in those subjects who received 200 and 400 mg doses. The most common adverse effect was nausea, which occurred in 6% of subjects who received 200 mg and 19% of those who received 400 mg. The study demonstrates the clinical application of oral CAmpB; however, further multi-dose evaluation and phase II clinical studies will provide further insights into the reality of this formulation. In a recent phase I ascending-dose study (*n* = 9/cohort), CAmpB containing 1 g, 1.5 g, or 2.0 g/day (divided doses) was administered to HIV survivors of cryptococcosis for three days, and a second seven-day trial (*n* = 9) at 1.5 g/day further assessed the tolerability [62,63]. All subjects completed their full dose without vomiting or experiencing other significant side effects, demonstrating tolerability to the subjects. However, the 1 g/day cohort experienced four mild adverse events, while the 1.5 g and 2.0 g cohorts experienced 7 and 20 mild adverse events, respectively. A qualitative survey revealed that 96% of the subjects preferred oral CAmpB dosing to the previously administered IV AmpB. The 1.5 g/day cohort completed 98% doses with only five minor adverse events and no evidence of nephrotoxicity. The study concluded that oral CAmpB is well tolerated at 4–6 doses per day without significant adverse effects. More recently, further progress by Martina Biopharma on their CAmpB (MAT2203) oral formulation involved a phase II trial with 100 HIV patients with cryptococcal meningitis treated with MAT2203 (CAmpB) in four cohorts. The first two cohorts received (MAT2203), with approval to proceed with another two cohorts [63]. The first two cohorts received MAT2203 as a follow-up therapy to IV, while the remaining two received monotherapy. All patients underwent a 14-day induction and four-week maintenance. The clearance rate of cryptococcus from cerebrospinal fluid (CSF) was the primary efficacy endpoint for this study. The overall survival rate in cohort 2 was 95% in 40 randomly assigned patients, and all patients who completed the treatment (MAT2203 induction/maintenance) achieved a sterile CSF culture. Furthermore, no patients developed new or relapsed cryptococcal infections during the treatment period, and both cohorts did not show evidence of renal toxicity or electrolyte imbalances.

Another multicenter randomized phase II study on oral CAmpB evaluated the tolerability and efficacy of 200 mg and 400 mg of CAmpB against fluconazole 150 mg in vulvovaginal candidiasis (VVC) [64]. Seventy-five patients with varying severity of VVC were divided into three groups and administered 200 mg or 400 mg CAmpB and 150 mg fluconazole. After a 12-day treatment period, clinical cure rates for the three cohorts were 52%, 55%, and 75%, respectively. In addition, the mycological eradication rates were 36%, 32%, and 85%, respectively. The sample size was determined empirically and was acceptable for this proof of concept. An oral AmpB would have been unthinkable just a few years ago. Although it took several years of intensive research, these clinical advances paved the way for the future development of a clinically viable oral AmpB formulation.

## 5. Conclusions

From the foregoing, it is apparent that there is a need for viable AmpB formulations free from manifestation of side effects associated with current IV formulations. The oral route provides an option for the delivery of AmpB but also presents several physiological and anatomical challenges that need to be surmounted in order to achieve clinically relevant plasma concentrations. Nanoformulation technology provides a platform to address these challenges. This review captures key formulation-related nanotechnological factors and physicochemical factors of nanocarriers that need to be considered when developing oral formulations of AmpB, thus contributing to improved systemic bioavailability. Clinical data on current and past studies reveal that a breakthrough for a viable oral AmpB nanoformulation is on the horizon; however, there appears to be a gap between the work done by formulation scientists and clinicians, which needs to narrow and intensify. 

## Figures and Tables

**Figure 1 pharmaceutics-14-01823-f001:**
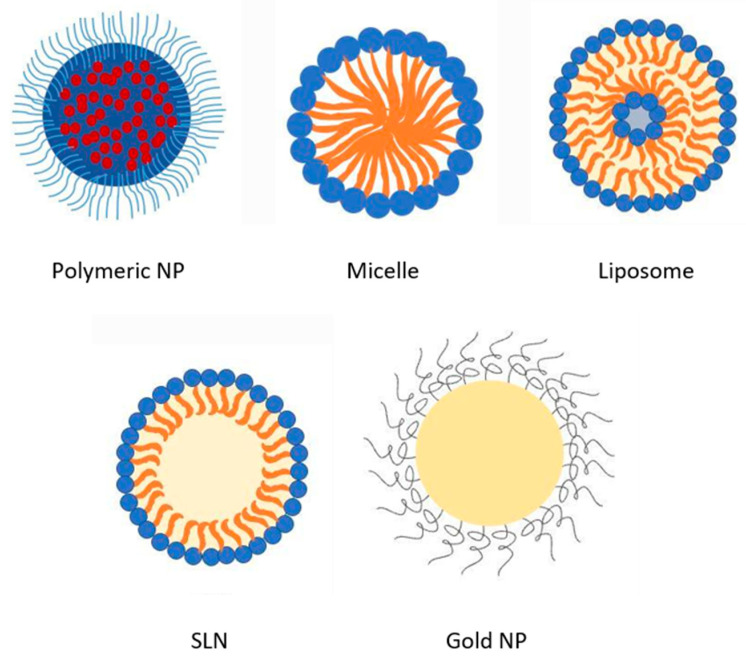
Nanoformulations under investigation for the oral delivery of AmpB.

**Figure 2 pharmaceutics-14-01823-f002:**
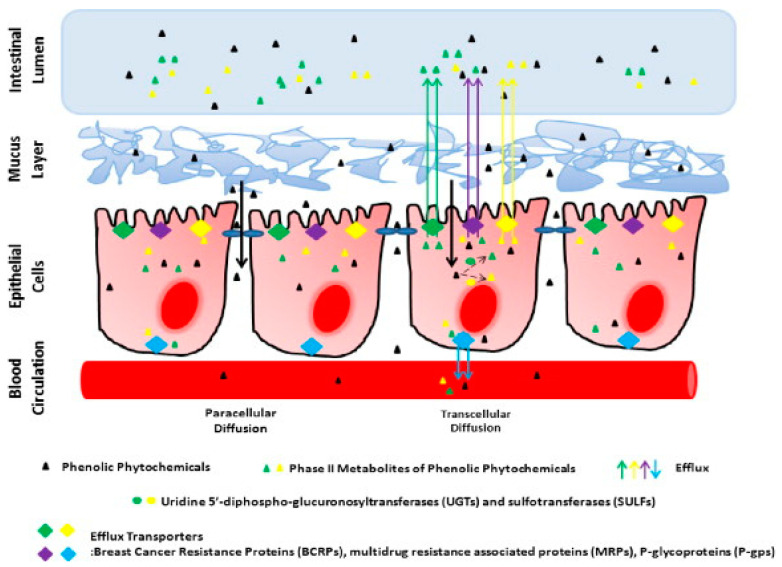
Process of passive diffusion of phenolic phytochemicals into epithelial cells followed by metabolism and active efflux. Adapted with permission from [49] 2022, Zheng et al.

**Figure 3 pharmaceutics-14-01823-f003:**
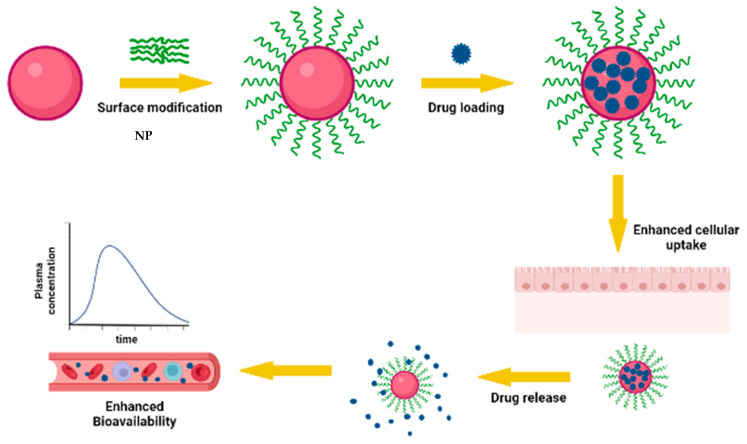
Enhanced bioavailability of AmpB nanoformulation offered by surface modifications.

**Table 1 pharmaceutics-14-01823-t001:** Summary of in vitro and in vivo studies on different types of oral AmpB nanoformulations.

Drug Delivery System	Method of Preparation, Major Components and Key Features	Outcome	Ref.
Nanoparticles Nanosuspensions	Nanoprecipitation: PLGA, poloxamer 188 and 388. Size: 86–153 nm; ZP −31.0 mV. Solvent-antisolvent precipitation: PVA, DMSO. Size: 118–400 nm; ZP −20 mV.	AMP-B-loaded NP showed 2-fold and AmpB nanosuspensions showed 4-fold enhancement in antifungal activity compared to AmBisome^®^ in a mouse model.	[35]
Mannose anchored thiomer nanocarriers	Covalent linkage of thioglycolic-chitosan followed by mannose addition: Mannose, chitosan. Size: 430–482 nm	6-fold increase in oral BA and 3-fold increase in half-life with significantly less toxicity than the AmpB control.	[36]
Liposomes	Modified injection method: Egg yolk phosphatidylcholine, cholesterol, ceramides. Size: 200 nm, EE > 75%	In an in vitro stomach–duodenum model, ceramides offered better membrane stability to ceramics anchored liposomes. Moreover, ceramides inhibited the detergent effect of bile salts on liposome membranes.	[37]
Cubosomes	High-pressure homogenization: Glyceryl monoolein, poloxamer 407. Size: 192 nm, EE > 94%	AmpB-loaded glyceryl monoolein cubosomes showed enhanced therapeutic efficacy than Fungizone^®^. A two-day treatment of 10 mg/kg dose was sufficient to attain therapeutic concentrations at the renal tissues for fungal treatment.	[38]
Solid lipid nanoparticles (SLN)	Nanoprecipitation followed by probe sonication: Glyceride dilaurate, phosphatidylcholine, PEG-660–12 hydroxystearate. Size: 200 nm, EE > 95%	In vivo pharmacokinetics evaluation on Wistar rats showed faster onset of action and prolonged half-life than pure drug solution.	[30]
Nanostructured lipid carriers (NLC)	Homogenization ultrasonication: Chitosan, beeswax, coconut oil. Size: 394 nm, EE > 86%	NLC formulation showed comparable antifungal (in vitro) efficacy than AmpB and is twice less toxic to RBCs.	[39]
Stealth nanoparticles	Emulsification diffusion: PLGA-PEG copolymers, PVA, vitamin E, pluronic F68. Size: <1000 nm, EE > 56%	PEG concentration plays a significant role in size, EE, and drug release. 15% PEG demonstrated a controlled drug release of 54% up to 24 h.	[40]
Nanocapsules	Nanoemulsion production by emulsion solvent evaporation followed by chitosan deposition. Mannose sugar, chitosan, soya lecithin, polysorbate 80. Size:198 nm; ZP +31 mV, EE 96%.	Increased macrophage selectivity by interacting with overexpressed mannose receptors, resulting in 90% reduction in spleen parasite load and decreased nephrotoxicity.	[41]
Lipopolymerosome	Single-step nanoprecipitation: Glycol–chitosan, stearic acid, soya lecithin, cholesterol. Size: 243 nm; ZP +27 mV.	In vitro and in vivo evaluation demonstrated improved plasma drug stability and reduced toxicity compared to AmBisome^®^ and Fungizone^®^. Enhanced anti-leishmanial activities and the glycol–chitosan copolymer was vital in improving the drug stability.	[42]
Polymer–lipid hybrid nanoparticles (PLN)	Desolvation method: Gelatin, lecithin, acetone, DMSO. Size: 253 nm, EE > 50.0%	Drug release followed Huguchi kinetics. Moreover, a 6-fold increase in intestinal permeability on Caco-2 cell lines and a 5-fold enhancement in oral BA was found compared to free AmpB.	[34]
Nanocochleates	Film hydration: Phosphatidylserine, lecithin, cholesterol, vitamin E. ZP −9 to −16 mV, EE > 50.0%.	Enhanced gastric stability and slow release in the GI medium. A confocal microscopy study demonstrated the integration of phosphatidylserine to the Caco-2 intestinal cell layers and slow drug release.	[43]
Self-emulsifying drug delivery systems (SEDDS)	Solvent evaporation: Glyceryl mono-oleate, PEG, phospholipids. Size: 200–400 nm, enhanced solubility stability in SGF or SIF	SEDDS significantly decreased fungal CFU in Sprague–Dawley rats infected with *A. fumigatus* and *Candida albicans* without causing renal toxicities.	[44]
Polymeric micelles	Solvent-diffusion and microfluidics technique: Copolymer Soluplus^®^ (Polyvinyl caprolactam–PVA–PEG), VitE-TPGS Size: 80 nm, EE 95.0%	Enhanced cell uptake (6-fold) and permeability (2-fold) in Caco-2 cells in vitro while being less toxic than free AmpB.	[45]
Nanofibers	Electrospinning technique. PLGA, chloroform, 2,2,2-trifluoroethanol. Size: 582 nm, controlled delivery of AmpB for 8 days.	For vulvovaginal candidiasis, the vaginal fungal load in a murine model was eliminated after three days of local treatment.	[46]

Abbreviations: PLGA, poly(lactic-co-glycolic acid); DMSO, dimethyl sulfoxide; PVA, poly vinyl alcohol; PEG, polyethylene glycol; CFU, colony-forming unit.

## Data Availability

Not applicable.

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
