# Peer review of "Orally Administered Amphotericin B Nanoformulations: Physical Properties of Nanoparticle Carriers on Bioavailability and Clinical Relevance"

_pharmaceutics, 2022, doi:10.3390/pharmaceutics14091823_

Round 1
Reviewer 1 Report
1. Authors must add one para about the drawbacks of Amphotericin nanoformulation.
2. How the highly poorly soluble drugs solubility can be enhanced.
3. Is there any change reported after encapsulating into nanoformulation.
4. Add about the drug loading, encapsulation and release data of AmB in Table.
5. The stability data of the nanoformulation must be elaborated. How it has been improved after encapsulating into nanoformulation.
Author Response
Reviewer 1
1. Authors must add one para about the drawbacks of Amphotericin nanoformulation.
Author’s response: Chapter 4 captures clinical trials involving amphotericin B nanoformulations, where relevant drawbacks have be presented. Also in section 3.1, whilst describing factors affecting bioavailability of orally administered amphotericin B nanoformulatiobn, key drawbacks are captured.
2. How the highly poorly soluble drugs solubility can be enhanced.
Author’s response: The factors affecting the solubility of amphotericin B is described in section 3.
3. Is there any change reported after encapsulating into nanoformulation?
Author’s response: The effect of encapsulation efficiency of on the bioavailability of amphotericin B is covered in section 3.2 line 311 onwards.
4. Add about the drug loading, encapsulation and release data of AmB in Table.
Author’s response: Authors believe that there will be a redundancy to include a table of the above data because they have been described in detail in the text. Furthermore, it is challenging to present numerical data on release in a table form.
5. The stability data of the nanoformulation must be elaborated. How it has been improved after encapsulating into nanoformulation.
Author’s response: Additional data on stability is now included in section 3.4, line 372-400 (blue fonts) onwards.
Reviewer 2 Report
[Pharmaceutics] Manuscript ID: pharmaceutics-1856540-peer-review-v1
Review 1
Orally Administered Amphotericin B Nanoformulations: Phys-2 ical Properties of Nanoparticle Carriers on Bioavailability and 3 Clinical Relevance”
Dear Authors,
The manuscript has a good review of the types of nanoformulations. Tables contain appropriate summarized information. Also, it has been tried to investigate the effect of several main variables, including the type of nanocarrier, the type of polymer and lipid used in the structure of nanoparticles, surface modification, stability, and encapsulation efficiency on bioavailability. It contains good information for researchers who work on the formulation of class IV drugs according to the BCS classification and has a good basis for choosing materials and formulations.
Comments:
1. Lin 42, Instead of word slow, it is better to mention the total time of IV administered.
“(0.5 - 1.5 mg/kg) mg/kg administered by slow intravenous infusion over 4 to 6 hours”
2. Lin 111, “The release of AmpB from polymeric nanocarriers is mostly preceded by desorption or diffusion from the polymer coating or matrix, respectively”
3. the release mechanisms of the drug in nanotherapeutic delivery systems will have some kinds of different mechanisms like desorption or diffusion or erosion. Considering that the release mechanism of nanocarriers plays an essential role in bioavailability, side effects, and stability in stomach conditions, it is better to add some explanations to the text. (Review A review of drug release mechanisms from nanocarrier systems, Chizhu Ding , Zibiao Li, Materials Science and Engineering C 76 (2017) 1440–1453), This article is suggested.
4. Line 132, “nanoparticles” is not appropriate phereses “liposome” is more accurate.
5. Line 136, Please mention the kind of system of drug delivery of “conventional AmpB deoxycholate”
6. Line 139, please explain clearer the role of the size of the liposome in lowering glomerular filtration.
7. Line 150 “Lipids employed in SLNs are affordable and safer than those used in other lipid nanoparticles”, please mention the reference.
8. line 169, Electrospun gelatin nanofibers are not classified as nanoparticles, nanosystem or nanoformulation for this system is more accurate.
9. Table 1:” pharmacokinetic outcomes” is more exact than “Out com”
10. Table 1: It is suggested to include nanofibers in the table as an example.
11. Table1: Abbreviations to the table should be noted in full. eg. DMSO, CFU
12. Line 242: “PLCNPs” should be “LCNPs” please check it.
13. Some sentences are repeated in the text, so it is better to remove the repetition. For example, Lin 223 and line 273, are the same.
14. Figure 2 is not mentioned in the text.
15. In section 3.4, For the relationship between drug release and drug stability, it is better to add some explanations.
16. If possible, please refer to the type of formulations used in Clinical studies in section 4.
17. Considering that in several places of the manuscript, the studied forms are compared with the commercial forms of Amphotericin B available in the market, if possible, please include the type of drug delivery system of the amphotericin brands in the text.
Regards,
Author Response
1.Lin 42, Instead of word slow, it is better to mention the total time of IV administered.
“(0.5 - 1.5 mg/kg) mg/kg administered by slow intravenous infusion over 4 to 6 hours”
Author’s response: The sentence has now been corrected as per reviewer’s suggestion (Line 44-45)
2. Lin 111, “The release of AmpB from polymeric nanocarriers is mostly preceded by desorption or diffusion from the polymer coating or matrix, respectively”
3. The release mechanisms of the drug in nanotherapeutic delivery systems will have some kinds of different mechanisms like desorption or diffusion or erosion. Considering that the release mechanism of nanocarriers plays an essential role in bioavailability, side effects, and stability in stomach conditions, it is better to add some explanations to the text. (Review A review of drug release mechanisms from nanocarrier systems, Chizhu Ding , Zibiao Li, Materials Science and Engineering C 76 (2017) 1440–1453), This article is suggested.
Author’s response for comments 2&3. Additional mechanisms on the release mechanisms from nanoformulations have now been included. (line 138-148).
4. Line 132, “nanoparticles” isnot appropriate phereses “liposome” is more accurate.
Author’s response: Reviewer suggestion has been adopted (line 167)
5. Line 136, Please mention the kind of system of drug delivery of “conventional AmpB deoxycholate”
Author’s response: Corrected as per reviewer’s suggestion (line 171)
6. Line 139, please explain clearer the role of the size of the liposome in lowering glomerular filtration.
Author’s response: An explanation for lowering of the glomerular filtration due to the size of the liposome has been now been included the revised manuscript, however this appears earlier than the statement in line 139 (line 82-93).
7. Line 150 “Lipids employed in SLNs are affordable and safer than those used in other lipid nanoparticles”, please mention the reference.
Author’s response: A reference has now been included in the revision (line 186)
8. line 169, Electrospun gelatin nanofibers are not classified as nanoparticles, nanosystem or nanoformulation for this system is more accurate.
Author’s response: Reviewer suggestion has been adopted (line 202).
9. Table 1:” pharmacokinetic outcomes” is more exact than “Out com”
Author’s response: Table corrected as per reviewer’s suggestion
10. Table 1: It is suggested to include nanofibers in the table as an example.
Author’s response: Corrected as per reviewer’s suggestion
11. Table1: Abbreviations to the table should be noted in full. eg. DMSO, CFU
Author’s response: Included the full-form of abbreviations at the bottom of the table
12. Line 242: “PLCNPs” should be “LCNPs” please check it.
Author’s response: Corrected as per reviewer’s suggestion (line 270)
13. Some sentences are repeated in the text, so it is better to remove the repetition. For example, Lin 223 and line 273, are the same.
Author’s response: Line 223 is removed in the amended version
14. Figure 2 is not mentioned in the text.
Author’s response: The figure is now captured in the text.
- In section 3.4, For the relationship between drug release and drug stability, it is better to add some explanations.
Author’s response: Further explanation is added to section 3.4. (in blue font)
- If possible, please refer to the type of formulations used in Clinical studies in section 4.
Author’s response- Although not all clinical studies include the formulation types, those that provide the information have been included in the section.
- Considering that in several places of the manuscript, the studied forms are compared with the commercial forms of Amphotericin B available in the market, if possible, please include the type of drug delivery system of the amphotericin brands in the text.
Author’s response- Liposomal amphotericin B (AmBisome®) and Amphotericin B infusion (Fungizone) are included in the text.(Introduction and Table)
Reviewer 3 Report
Dear Authors,
The manuscript ID: pharmaceutics-1856540_v1 entitled „Orally Administered Amphotericin B Nanoformulations: Physical Properties of Nanoparticle Carriers on Bioavailability and Clinical Relevance” written by Shadreen Fairuz, Rajesh Sreedharan Nair and Nashiru Billa is interesting.
Fungal infections are a major contributor to infectious disease-related deaths across the globe. Currently, there are only three major classes of drugs approved for the treatment of fungal infections and their efficacy is compromised by the development of drug resistance. Because of the increasing prevalence and changing microbiological spectrum of invasive fungal infections, some form of amphotericin B still provides the most reliable and broad spectrum therapeutic alternative. However, the use of amphotericin B deoxycholate is accompanied by dose-limited toxicities, most importantly, infusion-related reactions and nephrotoxicity. Therefore, appropriate solutions, such as nanoformulation technology, are needed. The Authors presented a comprehensive review that investigates the applications of nanotechnology in the development of oral Amphotericin B formulations. In my opinion, the topic of the work is current. The whole manuscript is properly organized and well written. Extensive literature data was used and adequate conclusions were drawn.
I have some suggestions in order to improve paper, which are the following:
1) References – please revise according to the instructions for the Authors (some bibliographic entries);
2) Other:
Line 40: spp, – spp. (without italics)
Lines 40, 56: and – without italics
Line 442: tclinicians – clinicians
Table 1: in vitro – italics; in vivo – italics
I think that this review is valuable and worth publishing in “Pharmaceutics”.
With highest regards,

Author Response
1. References – please revise according to the instructions for the Authors(some bibliographic entries);
Author’s response: References have now been checked and cited in MDPI style using endnote.
2. Other:
Line 40: spp, – spp. (without italics)
Lines 40, 56: and – without italics
Line 442: tclinicians – clinicians
Table 1: in vitro – italics; in vivo – italics
Author’s response- Reviewer’s comments corrected in the text
Reviewer 4 Report
Overall,, the topic is very interesting a dn worthy of investigation. However, the literature review has not been performed sistematically and hence, conclusions are limited. I suggested the authors include the following revisions in their manuscript:
1. I think it would be interesting if authors include a section indicating the mesh words utilised to performe this literature review.
2. It has not been discussed the aggregation state of the amphotericin B whihc actually plays a crucial role both in efficacy and toxicity as well as oral pharmacokinetics. The lost of patents also is important to discuss. Have a look to this paper: Amphotericin B formulations–the possibility of generic competition
3. Only polymeric nanoparticle based on PLGA have beenincluded in this work, but not carbohydrate based nanoparticles. Authors shoudl look carefully for other work performed on chitosan derivatives nanoparticles whihc has shown promising results. There are several articles on this have a look to this one for example: Oral particle uptake and organ targeting drives the activity of amphotericin B nanoparticles
4. Also, it is key that authors focus on whuch diseases the oral amphotericin B can be potentially useful. Fungal infections such as candida and aspergillosis, but also parasitic infections such as leishmaniasis and trypanosomiasis. There are papers paublished already about oral amphotericin B nanomedicines for amercian trypanosomiasis showing its efficacy. Check this paper: Engineering Oral and Parenteral Amorphous Amphotericin B Formulations against Experimental Trypanosoma cruzi Infections
5. It is important that authors understan where the parasites or fungus are accumulated to correlate these values with the concentrations acehieved in tissues and hence, the efficacy. There is a good review already published in this topic which will help you to organise and complete your information: Oral amphotericin B: The journey from bench to market
Author Response
1. I think it would be interesting if authors include a section indicating the mesh words utilised to perform this literature review.
Author’s response: Mesh words Included in the introduction section (lines 120-125)
2. It has not been discussed the aggregation state of the amphotericin B whihc actually plays a crucial role both in efficacy and toxicity as well as oral pharmacokinetics. The lost of patents also is important to discuss. Have a look to this paper: Amphotericin B formulations–the possibility of generic competition
Author’s response: The effect of the aggregation state on toxicity is now added (line 80-86)
3. Only polymeric nanoparticle based on PLGA have been included in this work, but not carbohydrate based nanoparticles. Authors should look carefully for other work performed on chitosan derivatives nanoparticles whihc has shown promising results. There are several articles on this have a look to this one for example: Oral particle uptake and organ targeting drives the activity of amphotericin B nanoparticles
Author’s response: Additional material has now been added to reflect reviewer suggestion (Line 168-171)
4. Also, it is key that authors focus on whuch diseases the oral amphotericin B can be potentially useful. Fungal infections such as candida and aspergillosis, but also parasitic infections such as leishmaniasis and trypanosomiasis. There are papers paublished already about oral amphotericin B nanomedicines for amercian trypanosomiasis showing its efficacy. Check this paper: Engineering Oral and Parenteral Amorphous Amphotericin B Formulations against Experimental Trypanosoma cruziInfections
Author’s response: Included in section 1 as per the reviewer’s suggestion (line 40-42)
5. It is important that authors understand where the parasites or fungus are accumulated to correlate these values with the concentrations acehieved in tissues and hence, the efficacy. There is a good review already published in this topic which will help you to organise and complete your information: Oral amphotericin B: The journey from bench to market
Author’s response: Information has been included in section 1 as per the reviewer’s suggestion. (line 47-48)
Round 2
Reviewer 1 Report
accept
Author Response
Comments accepted.
Reviewer 4 Report
I cannto see the corrections made by the authors or at least the lines that authors state in the replay letter were changes have been made do not correlate to check.
Author Response
I think it would be interesting if authors include a section indicating the mesh words utilised to perform this literature review.
Author’s response: Mesh words Included in the introduction section (lines 125-131)
2. It has not been discussed the aggregation state of the amphotericin B whihc actually plays a crucial role both in efficacy and toxicity as well as oral pharmacokinetics. The lost of patents also is important to discuss. Have a look to this paper: Amphotericin B formulations–the possibility of generic competition
Author’s response: The effect of the aggregation state on toxicity is now added (line 81-86)
3.Only polymeric nanoparticle based on PLGA have been included in this work, but not carbohydrate based nanoparticles. Authors should look carefully for other work performed on chitosan derivatives nanoparticles whihc has shown promising results. There are several articles on this have a look to this one for example: Oral particle uptake and organ targeting drives the activity of amphotericin B nanoparticles
Author’s response: Additional material has now been added to reflect reviewer suggestion (Line 168-171)
Also, it is key that authors focus on whuch diseases the oral amphotericin B can be potentially useful. Fungal infections such as candida and aspergillosis, but also parasitic infections such as leishmaniasis and trypanosomiasis. There are papers paublished already about oral amphotericin B nanomedicines for amercian trypanosomiasis showing its efficacy. Check this paper: Engineering Oral and Parenteral Amorphous Amphotericin B Formulations against Experimental Trypanosoma cruziInfections
Author’s response: Included in section 1 as per the reviewer’s suggestion (line 40-45)
It is important that authors understand where the parasites or fungus are accumulated to correlate these values with the concentrations acehieved in tissues and hence, the efficacy. There is a good review already published in this topic which will help you to organise and complete your information: Oral amphotericin B: The journey from bench to market
Author’s response: Information has been included in section 1 as per the reviewer’s suggestion. (line 47-48)
Reviewer 4 (2nd round)
I cannto see the corrections made by the authors or at least the lines that authors state in the replay letter were changes have been made do not correlate to check.
Author’s response. Response: The line numbers had moved slightly after adding some texts. Correct line numbers are now included in the responses.
Round 3
Reviewer 4 Report
Authors have addressed properly the comments suggested by reviewers so manuscrito is ready for publication.